# The Impact of the Foliar Application of Amino Acid Aqueous Fertilizer on the Flavor of Potato Tubers

**DOI:** 10.3390/foods12213951

**Published:** 2023-10-29

**Authors:** Songhe Hu, Kaifeng Li, Xing Zhang, Changwei Yang, Rui Zhang, Huachun Guo

**Affiliations:** 1College of Agronomy and Biotechnology, Yunnan Agricultural University, Kunming 650201, China; hu_0208@163.com (S.H.); dtllx04@163.com (K.L.); bryan960614@163.com (X.Z.); 18184809241@163.com (C.Y.); zhang134rui@163.com (R.Z.); 2Yunnan Engineering Research Center of Tuber and Root Crop Bio-Breeding and Healthy Seed Propagation, Yunnan Agricultural University, Kunming 650201, China; 3Tuber and Root Crop Institute, Yunnan Agricultural University, Kunming 650201, China

**Keywords:** potato, tuber, amino acid aqueous fertilizer, volatile flavor compounds, free amino acids

## Abstract

The quality of the flavor of potatoes is a crucial determinant in the commercial success of a potato variety. Plant nutrition promotes the synthesis of amino acids, and the application of exogenous amino acids has the potential to enhance the flavor quality of potatoes. In this experiment, Dian Shu 1418 and Dian Shu 23 were used as the materials, and different amino acid foliar spray trials were designed. The free amino acid content in potato tubers is determined based on high-performance liquid chromatography, and volatile tuber flavor compounds are detected using gas chromatography–mass spectrometry. The results showed that the amino acid foliar spray effectively increased the content of glycine, methionine, and phenylalanine in the tubers, subsequently increasing the levels of 2,3-dimethyl-pyrazine and 2-ethyl-3-methyl-pyrazine, enhancing the roasted fragrance characteristics of the potatoes. The experimental results provide a reference for cultivating flavor enhancement in potato tubers.

## 1. Introduction

As is well known, potato (*Solanum tuberosum* L.) has always been an important food crop. With the rapid development of the economy, people are now pursuing a more delicious and healthy way of eating potatoes, so many potatoes are consumed as fresh produce [1,2]. With the development of the economy, people’s consumption level continues to increase. The demand for potato quality is gradually increasing, and potato flavor is one of the most important factors in determining quality and consumption preference [3].

Potato flavor represents a combination of taste, aroma, and texture, perceived through taste and olfactory senses [4]. The primary constituents of flavor are volatile compounds, including lipids, aldehydes, alcohols, ketones, acids, esters, hydrocarbons, amines, furans, pyrazines, and sulfur compounds [5]. During cooking, volatile flavor compounds are formed from precursor substances through enzyme-catalyzed and non-enzyme-catalyzed reactions. These precursors are stored in raw potato tubers and comprise sugar, amino acids, nucleotides, and lipids. The content of these compounds and the enzymes involved in their transformation into flavor compounds vary considerably between different genotypes, production environments, and storage conditions, and different cooking methods yield varying types and amounts of volatile flavor compounds [5,6,7]. Therefore, the main measure to improve potato flavor quality is to improve the flavor precursors.

Amino acids are significant precursors of potato flavor, interacting with reducing sugars at high temperatures to undergo Maillard reactions, generating furan and pyrazine substances. These reactions have a strong positive correlation with the roasted flavor of potatoes, representing the most crucial and characteristic components in baked potato flavor [8,9,10]. The influence of amino acids on flavor has been reflected in many reports. Among them, the correlation analysis of the metabolic components in potato tubers and the flavor components after maturation found that three compounds, alanine, methionine, and sucrose, may be the main metabolic compounds that affect the formation of the characteristic potato flavor [11]. Employing genetic engineering to introduce cystathionine γ-synthase (CGS), a key enzyme in the biosynthesis of methionine in ‘Russet Burbank’ potatoes, can effectively increase soluble methionine content, thus enhancing the flavor stability and quality of processed potato products [12]. At present, the main agronomic measures are the application of various fertilizers. Nitrogen fertilization can significantly increase lysine and methionine amino acids [13], and the foliar application of amino acid aqueous fertilizer has a noticeable promoting effect on tuber yield and dry matter content [14]. However, there are only a few studies that attempt to indirectly improve flavor compounds by influencing the potato flavor precursors through agronomic measures.

This study employs various concentrations of amino acid foliar fertilizers as treatment methods, utilizing gas chromatography–mass spectrometry (GC-MS) to determine volatile flavor compounds and chromatographic methods for endogenous amino acid quantification. It comprehensively examines the changes in endogenous amino acids and volatile compounds at different treatment levels, aiming to elucidate the impact of exogenous amino acid application on endogenous synthesis and to explore the possibility of its indirect influence on the changes in flavor compound content, hoping to provide a theoretical basis for the cultivation of enhanced potato flavor.

## 2. Materials and Methods

### 2.1. Experiment Materials and Fertilizers

In this experiment, the Potato Research Institute of Yunnan Agricultural University provided two potato varieties, Dianshu 23 and Dianshu 1418. The fertilizer used for the experiment is an amino acid water-soluble fertilizer [15].

### 2.2. Reagents and Equipment

n-Alkanes (C7-C40) and 2-methyl-3-heptanone (0.163 µg·µL^−1^) are chromatographically pure standard reference materials that were purchased from the Sigma-Aldrich chemical company (St. Louis, MI, USA). The equipment used includes a gas chromatography mass spectrometry (GCMS) instrument (model 7890A/5970B, Agilent Technologies, Santa Clara, CA, USA), an amino acid analyzer (Biochrom 30+ series, Biochrom Ltd. (Cambridge, UK), CBG), and an electric air fryer (Joyoung Co., Ltd. (Jinan, China), HGH).

### 2.3. Experimental Method

This experiment designed three foliar fertilizer concentration treatments based on conventional fertilization: foliar spray with clear water (CK), foliar spray with 0.3% concentration of amino acid water-soluble fertilizer (AA_0.3), and foliar spray with 0.6% concentration of amino acid water-soluble fertilizer (AA_0.6), with four repetitions for each treatment. Clearwater spraying was used as a control. The foliar spray was applied to the test varieties during the potato budding stage (50–60 days after sprouting), with a spraying interval of 7 days, and 4 sprays until the leaves were covered with water droplets but not dripping. The trial adopted the field plot experimental method with a randomized block design. The plant row spacing was 65 cm × 25 cm, with 8 plants per row, and each plot contained 4 rows, planting 32 plants per plot. Each plot measured 5 m² (2.5 m × 2 m), and a 1.1-m-wide protective row was established around the experimental field.

### 2.4. Sampling and Index Measurement

Individual plants were harvested at the time of potato tuber maturity, with 10 fresh potato tubers randomly selected from each plot. The fresh potato tubers were peeled and cut into small pieces of 1 cm × 1 cm × 1 cm. The pieces were frozen in liquid nitrogen, with 20 g taken for freeze drying, and ground into powder for free amino acid measurement, following the same procedures as in the study by Cieślik [16], and specific amino acid quantitative methods refer to “the determination of amino acids in foods-GB 5009.124–2016”. Another 20 g of fresh potato tubers were placed into a preheated air fryer and baked at 200 °C for 18 min. Then, 2 g samples were weighed and put into a headspace bottle for the determination of volatile flavor substances. For the specific operation details and quantitative data methods, refer to the experiment of Li K [17]. The experiment calculated the ROAV based on the measured volatile flavor compound contents, referring to threshold data provided by Gemert and others [18], using the formula
(1)$${\mathrm{ROAV}}_{x}\approx 100\times \left({C\%}_{x}\times T_{\mathrm{stan}}\right)÷\left(C_{\mathrm{stan}}\times T_{x}\right)$$

as mentioned in the article by Wang Y [19], where *C*%*_x_* represents the relative percentage content of each volatile component, and *C*%_stan_ represents the relative percentage of components that contribute the most to the overall flavor of the sample; *T_x_* represents the sensory threshold corresponding to each volatile component, and *T*_stan_ represents the sensory threshold corresponding to the component that contributes the most to the overall flavor of the sample. The aroma characteristics mainly refer to the research literature by authors like Angeloni, CHEN, and others [20,21,22,23,24,25].

### 2.5. Data Processing and Statistical Analysis

For the evaluation data under different fertilization treatments, the test data were sorted by Microsoft Excel (2021) software, and the statistical software package based on R (version 4.3.1) was used for T-test data normalization, PCA, and cluster analysis. The contents of volatile flavor compounds and amino acids were determined first, and an FDR correction was performed. The free amino acids and volatile compounds were then normalized, followed by the screening of critical substances using the “ggraph” software package and Cytoscape 3.1 for the mapping of the relevant networks. The “FactoMineR” package was used to complete PCA rendering, and then the “factoextra” was used for the principal component analysis of all the screened data. For the evaluation data under different fertilization treatments, the test data were sorted by Microsoft Excel (2021) software, and the statistical software package based on R (version 4.3.1) was used for T-test data normalization, PCA, and cluster analysis. The contents of volatile flavor compounds and amino acids were determined first, and FDR correction was performed. The free amino acids and volatile compounds were then normalized, followed by the screening of critical substances using the “ggraph” software package and Cytoscape 3.1 for the mapping of relevant networks. The “FactoMineR” package was used to complete PCA rendering, and then the “factoextra” was used for the principal component analysis of all the screened data.

## 3. Results

### 3.1. The Impact of Foliar Fertilizer Spraying on the Free Amino Acid Content in Potato Tubers

In this study, a total of 42 amino acids and amino acid components were detected in this test. After eliminating the elements with less than a 50% detection rate, 22 amino acids and amino acid components, including aspartate, were obtained (Table 1), along with the comparative heat map of amino acid component contents (Figure 1). Different varieties of potatoes showed variations in the composition of the 22 free amino acids in the tubers after applying foliar fertilizer. Most of the amino acid components increased under the conditions of amino acid water-soluble fertilizer treatment, with only a few amino acids, including proline (Pro), ornithine (Orn), and citrulline (Citr), showing lower content in the Dian Shu 1418 variety compared to the control. A small portion of amino acids, such as asparagine (Asn) and glutamine (Gln), were lower in the Dian Shu 23 variety compared to the control. As we all know, aspartate and glutamine will produce acrylamide under high-temperature conditions; acrylamide is a recognized carcinogen; that is to say, after applying amino acid water-soluble fertilizer, Dian Shu 23 tubers are healthier after baking, which is an unexpected finding. The cluster analysis revealed that the amino acid water and fertilizer ratio at a 0.6% concentration benefitted the endogenous amino acid content of Dian Shu 23 treated with 0.3% AA. According to the heat map, it can be seen that the amino acid content changes in the potato tubers of the two varieties after treatment are not completely consistent, which may be caused by the variety effect. However, in the two varieties of potatoes, some endogenous amino acids have the same change trend, and these amino acids may be the focus of our attention. The results of cluster analysis showed that the concentration of water-soluble fertilizer with different amino acids had different effects on the content of free amino acids in tubers of different potato varieties. After clustering, it is obvious that potato varieties are taken as the difference. According to the heat map of this experiment, amino acid water and fertilizer treatment at a 0.6% concentration had the greatest effect on the endogenous amino acid content of Dian Shu 23. It is not difficult to compare the heat map with the endogenous amino acid content table. The contents of the endogenous amino acids of the Dian Shu 23 and Dian Shu 1418 potato varieties were significantly different under CK treatment. Table 1 shows that the free amino acid content in the tuber of Dian Shu 23 is much higher than the endogenous amino acid content of Dian Shu 1418, and most of the amino acid content of the two varieties shows an increasing trend under the treatment of water-soluble fertilizer. The results indicated that the application of water-soluble amino acid fertilizer on the leaf surface had a certain effect on the change in the contents of endogenous amino acid in potato tuber.

### 3.2. The Impact of Foliar Fertilizer Spraying on Volatile Flavor Compounds in Potato Tubers

Fresh potatoes undergo a series of Maillard reactions after baking in an air fryer, producing a range of volatile substances that affect the flavor of the potatoes. This study identified 89 volatile flavor compounds through GC-MS. After removing 15 components with a detection rate below 50%, this showed the contents of 74 volatile flavor compounds, including aldehydes, lipids, and pyrazines, and the contents comparison of 74 volatile flavor compounds can be seen in Table 2 (Figure 2). We found that the contents of most volatile substances in the tubers of Dian Shu 1418 and Dian Shu 23 significantly increased, whereas the contents of a few flammable substances decreased considerably under the treatment of water-soluble amino acid fertilizer with different concentrations. Due to varietal effects, the degree of increase (or decrease) of volatile flavor substances in the tubers of Dian Shu 1418 and Dian Shu 23 under amino acid water-soluble fertilizer treatment varied. As can be seen from Table 2, the contents of volatile substances produced by the two varieties are very different. Although the change amplitude is not consistent after the treatment of leaf fertilizer, the changing trend of most related substances is consistent, including some pyrazines, such as 2, 3-diethyl-5-methylpyrazines, which are clearly the main contributors that affect potato flavor. It can be shown that the measure of improving potato flavor by spraying amino acid water-soluble fertilizer on the leaf surface is effective. According to the cluster analysis in the heat map, the application of water-soluble fertilizer with different concentrations of amino acids had basically the same influence trend on the two different varieties. In terms of the influence of volatile substance content, the influence range of 0.3%_AA treatment is larger than that of 0.6_AA treatment, and the increase in volatile substance content is more obvious. However, according to the specific substances, some of the substances that increased significantly under a 0.3% concentration treatment are intermediate products that affect the main flavor substances of potatoes, indicating that the concentration selection of treatment has a great influence on the flavor of fried potato tubers in the study of the influence on potato flavor. In the clustering process, the growth or reduction degree of the synthesized volatile flavor substances was classified according to the concentration of water-soluble fertilizer treatment, and the 0.3% treatment and 0.6% concentration treatment were clearly separated, indicating that the application of amino acid foliar fertilizer needs concentration screening in order to better act on plants and ensure that the application of relatively less fertilizer can achieve better potato flavor improvements.

### 3.3. Analysis of the Correlation between Volatile Flavor Comp8onents and Free Amino Acids in Potato Tubers

Two varieties of potato tubers were subjected to a non-equilibrium condition T-test for the volatile compounds and the free amino acids detected within them. After screening and elimination, a total of 21 metabolites with significant differences (*p* < 0.05) were identified (Figure 3). These 21 metabolites include seven free amino acids, six pyrazine compounds, and eight other metabolites. These differentially occurring metabolites may influence the flavor of potatoes, and a joint correlation analysis of these 21 metabolites was conducted using the Parson method. The original data, which were log-transformed and normalized by the Pareto method using R language, were analyzed. The metabolic compounds shown in Figure 3 represent a significant correlation network (*p* < 0.05) with an absolute correlation coefficient greater than 0.65. Among the seven free amino acids shown in the figure, glycine (Gly) has the broadest correlation with volatile substances in potato tubers, being related to 11 substances, among which it has the highest correlation of 0.83 with 2-Acetylpyrrole (1-(1H-Pyrrol-2-yl)-ethanone). Next is Alanine (Ala), which is related to nine substances, having the highest correlation of 0.94 with 3-Ethyl-2,5-dimethyl-pyrazine. The other free amino acids in the figure include leucine (Leu), isoleucine (Ile), methionine (Met), phenylalanine (Phe), and valine (Val), all of which have a correlation of at least 0.65 with one or more volatile substances. Among the volatile flavor substances in the figure, seven have a correlation of above 0.85 with free amino acids, namely 1-(2-Furanylmethyl)-1H-pyrrole, 1-(1H-pyrrol-2-yl)-ethanone, 2-Methoxy-phenol, 2,3-Diethyl-5-methyl-pyrazine, 2-Ethyl-3-methyl-pyrazine, 3-Ethyl-2,5-dimethyl-pyrazine, and 3,5-Diethyl-2-methyl-pyrazine, six of which are correlated with Alanine. In the figure, (E, E)-2,6-nonadienal shows a negative correlation of 0.65 with five free amino acids: valine, phenylalanine, leucine, isoleucine, and methionine. Other volatile substances include 2-Isobutyl-3-methyl-pyrazine, (E, E)-2,4-Heptadienal, Benzaldehyde, Benzeneacetaldehyde, Furfural, and 2-Ethenyl-6-methyl-pyrazine, all of which have a correlation of at least 0.65 with one or more amino acids.

### 3.4. Effects of Foliar Amino Acid Fertilizer Spraying on the Main Free Amino Acid Content in Potato Tubers

Specifically, the seven amino acids closely related to the composition of volatile flavor compounds are glycine, leucine, isoleucine, valine, methionine, alanine, and phenylalanine. The treatment influences the variations in the content of each free amino acid and exhibits variety-specific effects (Figure 4). Overall, amino acid fertilizer spray has a significant (*p* < 0.05) impact on Dian Shu 23. According to the figure, the variations in the content of the seven free amino acids in potato tubers are as follows: 0.6_AA treatment > 0.3_AA treatment > CK, and the 0.6_AA treatment achieved a significant level (*p* < 0.05) compared to the control, indicating that, compared to the 0.3_AA treatment, the 0.6_AA treatment had a more pronounced effect on the endogenous amino acid content in potato tubers. Due to the limited reports on the concentration of amino acid fertilizer, further experimentation is needed to determine the optimal concentration of amino acid fertilizer.

### 3.5. ROAV Analysis of Potato Volatile Flavor Substances after the Foliar Application of Amino Acid Aqueous Fertilizer

According to the correlation analysis network graph, 14 volatile flavor substances were identified (Table 3), and their corresponding ROAV values were calculated based on the formula [19]. According to the data in Table 1, the substances mainly affecting the aroma of potatoes after amino acid aqueous fertilizer treatment are 3-Ethyl-2,5-dimethyl-pyrazine, 2-Ethyl-3-methyl-pyrazine, 2-Isobutyl-3-methyl-pyrazine, Dimethoxyphenol, and (E,E)-2,6-Nonadienal. In Table 1, 3-Ethyl-2,5-dimethyl-pyrazine, 2-Ethyl-3-methyl-pyrazine, and 2-Isobutyl-3-methyl-pyrazine show a similar trend of ROAV changes, which all significantly increased compared to CK after the treatment, especially under the 0.6_AA treatment. Conversely, the ROAV values of Dimethoxyphenol and (E,E)-2,6-Nonadienal decreased with the increase in the spraying concentration of the amino acid aqueous fertilizer, the fat flavor and other odors decreased, suggesting that the application of amino acid aqueous fertilizer has a positive effect on air-fried potatoes. It also showed that the baked potato flavor in potato tuber was enhanced with the increase in amino acid foliar fertilizer concentration.

### 3.6. Analysis of Principal Components of Main Volatile Substances and Free Amino Acids under Amino Acid Water-Soluble Fertilizer Treatment

Therefore, a principal component analysis was performed on each treatment based on the representative volatile compounds and free amino acids screened in the experiments. Figure 5 shows the principal component analysis of the main volatile compounds and free amino acids under two different amino acid foliar fertilizer applications for two varieties. As seen in the figure, each treatment can be separated from the control in the first and second dimensions, indicating that amino acid foliar fertilization caused changes in the intrinsic amino acid content in the potato tubers and indirectly affected the composition of volatile flavor substances. Specifically, when compared with the two varieties in Figure 5, the overall content of valine, isoleucine, and leucine in variety Dian Shu 23 was higher than that of Dian Shu 1418. Notably, the volatile compound content in Dian Shu 23 is lower than in Dian Shu 1418. The above changes reflect the complexity of the synthesis of potato flavor substances.

## 4. Discussion

Nutrients can be absorbed into plants through diffusion via roots or leaves. The absorption of nutrients through the root system requires active or passive transport to reach the leaves, making foliar fertilizer application a more direct and effective fertilization measure than root absorption [26].

Foliar fertilization in potato plants can enhance tuber yield and quality [27,28,29]. This study found that amino acid fertilizers sprayed on the leaves of potatoes can effectively increase the level of free amino acids in the tubers, thereby increasing the content of flavor compounds in potatoes after maturation.

Volatile compounds in tubers are mainly produced by precursors, such as amino acids and sugars, through a series of thermal chemical reactions, and the resulting aroma is affected by amino acid content [30]. Nooshkam and others found that the content of antioxidants can influence the synthesis of volatile compounds [31]. Therefore, the differences in secondary metabolite content and sugar levels between the two varieties may be the reason for the higher levels of volatile compounds in DianShu 1418 compared to DianShu 23. Principal component 1 accounted for 54% of the variation between the varieties, and the treatments with two different fertilizer concentrations were separated from the control in principal component 1. The levels of internal glycine, methionine, etc., increased with the increase in external application concentration. The components and content of free amino acids contained in potato tubers profoundly influence the flavor quality of potatoes [5].

Pyrazine compounds are characteristic flavor substances in baked potatoes. In 1998, Wagner and others found that 3-Ethyl-2,5-dimethyl-pyrazine is a major contributor to the flavor of chips and baked potatoes [32]. According to research by Pareles and others, a mixture of 2-Isobutyl-3-methyl-pyrazine, 2,3-Diethyl-5-methyl-pyrazine, and 3,5-Diethyl-2-methyl-pyrazine has more of a baked potato aroma than previous single compounds [24]. 2-Ethyl-3-methyl-pyrazine has a distinct roasted smell and flavor, resembling that of roasted cocoa or nuts [25]. The above pyrazine substances are significantly positively correlated with glycine and alanine, indicating that the content of these two amino acids greatly affects the production of most pyrazine compounds, and their presence is vital to the flavor characteristics of potatoes. According to previous research, glycine, as the main carbon skeleton of the Maillard reaction, has the most significant influence on potato flavor substances. At the same time, alanine is the nitrogen source substrate for forming pyrazine substances, so glycine and alanine are the main components in forming pyrazine [33,34].

Additionally, 2,3-Diethyl-5-methyl-pyrazine and 2-Ethyl-3-methyl-pyrazine have ROVA values greater than 1. The changes in the content of these compounds contribute to the main flavor characteristics; pyrazine compounds have roasted and nutty flavors, etc. Therefore, applying exogenous amino acid fertilizers might enhance the roasted fragrance of potato tubers. The roasted fragrance has a positive correlation with the overall fragrance quality. Therefore, using exogenous amino acids has promoted the overall improvement in potato flavor. Besides pyrazine compounds, this study used GC-MS to determine volatile flavor substances in tubers treated with foliar-sprayed amino acid water-soluble fertilizer after air frying, detecting 74 volatile flavor compounds, including aldehydes, lipids, pyrazines, etc.

Phenylacetaldehyde has a relatively low flavor dilution factor and is a key intermediate active carbonyl formed during the Maillard reaction. The flavor substance produced is benzaldehyde, which has a significant fragrance and is closely related to four precursor substances: valine, threonine, isoleucine, and glycine [35]. (E, E)-2,4-Heptadienal and (E, E)-2,6-Nonadienal have fragrances, but (E, E)-2,6-Nonadienal will produce an unpleasant smell when it exceeds the odor threshold. In this experiment, the content of (E, E)-2,6-Nonadienal significantly increased, showing a negative correlation with threonine, valine, leucine, and isoleucine. The Maillard reaction products have antioxidant properties [31], and compounds like (E, E)-2,6-Nonadienal are produced by linoleic acid oxidation [36]. Therefore, possibly in the active 0.6_AA treatment of the Maillard reaction, lipid oxidation was inhibited, eventually leading to a decrease in aliphatic aldehyde compounds, possibly reducing the generation of unpleasant flavors [37]. Research found that leucine and isoleucine are the main representatives of Strecker aldehydes in the Maillard reaction [38]. In potatoes, threonine, as a sulfur-containing amino acid, produces hydrogen sulfide and ammonia through Streeker degradation, becoming a precursor substance for generating heterocyclic compounds [39]. Moreover, this study found that 1-(2-furfurylmethyl)-1H-pyrrole, 2-acetylpyrrole, and 2-methoxyphenol are significantly positively correlated with glycine and alanine, implying that the above three substances might be closely related to potato flavor characteristics. 2-Methoxyphenol is significantly correlated with leucine, isoleucine, threonine, and valine, and the flavor characteristics of these four substances have not been reported yet; they might be intermediates of the Maillard reaction, and further exploration is needed.

## 5. Conclusions

After applying amino acid water-soluble fertilizer to potato leaves, the content of seven amino acids closely related to the composition of volatile flavor compounds in the tubers increased. These include glycine, leucine, isoleucine, valine, methionine, alanine, and phenylalanine. As the precursors of volatile flavor compounds, free amino acids mainly form pyrazine compounds through the Maillard reaction after air frying. The content of 2,3-Dimethyl-5-pyrazine and 2-Ethyl-3-pyrazine increased, enhancing the roasted fragrance characteristics of the potatoes. The experiment showed that these could improve the flavor substance content and, hence, the flavor quality of potato tubers by increasing the umami amino acid content in the tubers through the foliar application of amino acid water-soluble fertilizer.

## Figures and Tables

**Figure 1 foods-12-03951-f001:**
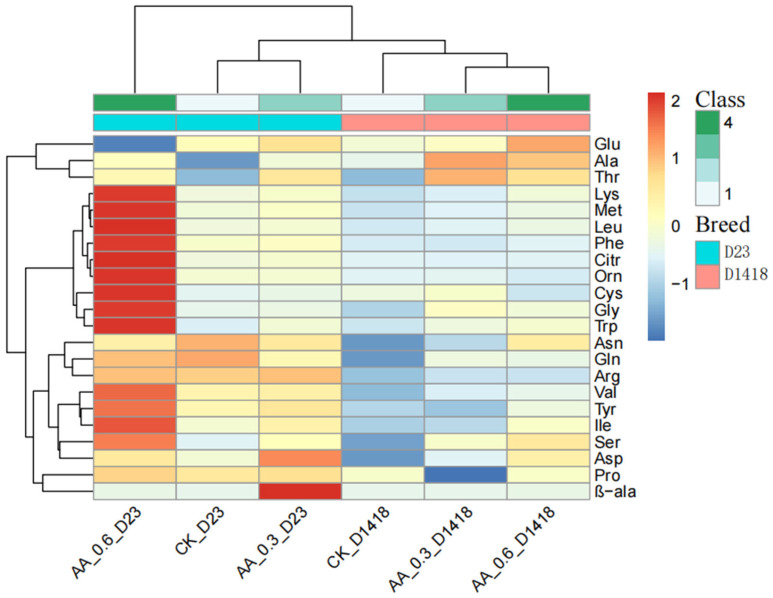
Comparison of free amino acid content in tubers of different potato varieties under amino acid water-soluble fertilizer treatment.

**Figure 2 foods-12-03951-f002:**
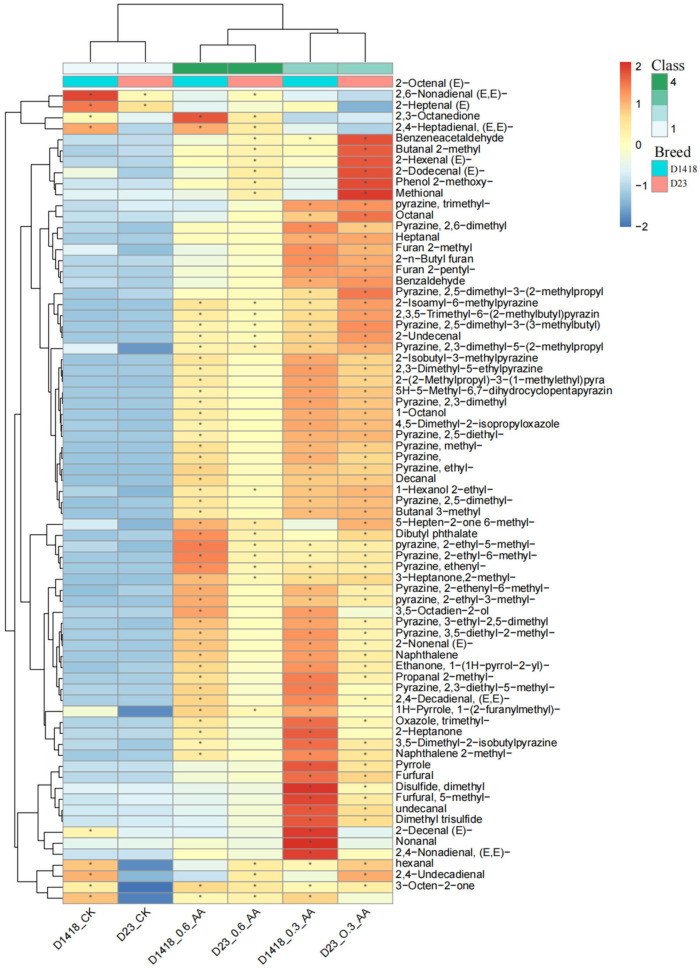
Comparison of volatile substances in potato tuber treated with amino acid concentration. (Note: ‘*’ means significant different at *p* < 0.05 among treatments).

**Figure 3 foods-12-03951-f003:**
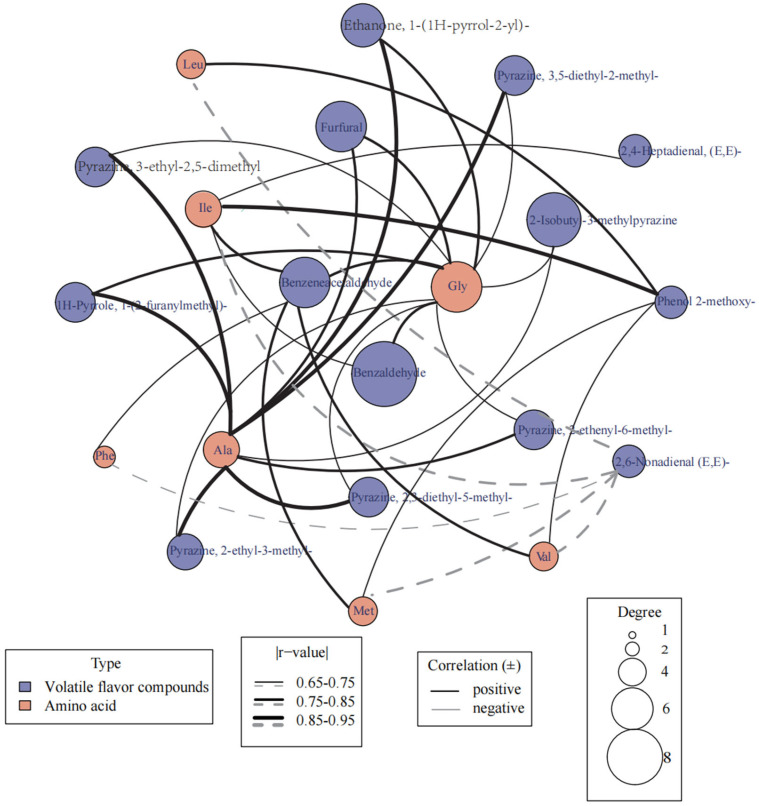
Correlation network diagram between volatile flavor components and free amino acids.

**Figure 4 foods-12-03951-f004:**
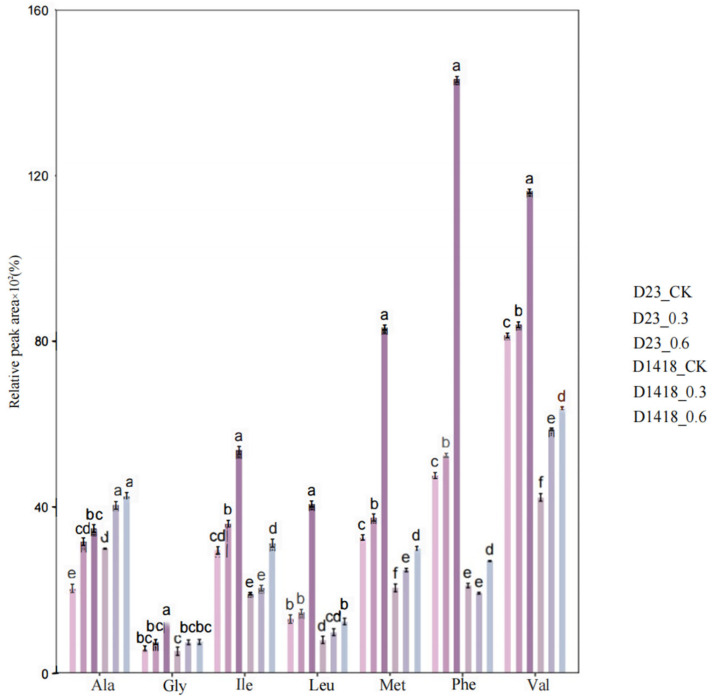
The effect of amino acid water-soluble fertilizer treatment on the amino acid content in the tubers of different potato varieties (lines). (Note: The significant difference among treatments indicated by different lowercase letters (*p* < 0.05)).

**Figure 5 foods-12-03951-f005:**
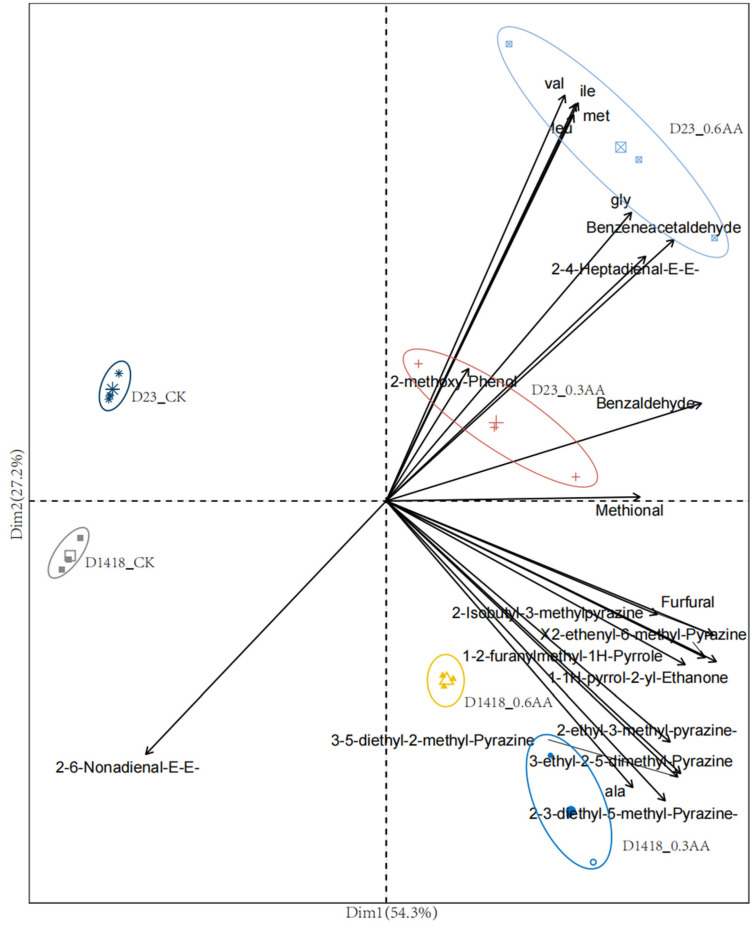
Principal component analysis of main volatile compounds and free amino acids under amino acid foliar fertilizer treatment. We have marked the different levels of processing represented by symbols of different colors.

**Table 1 foods-12-03951-t001:** Content of free amino acids in the tubers of different potato varieties treated with water-soluble fertilizer.

Name	D23 (ug·g^−1^)			D1418 (ug·g^−1^)		
	CK	AA_0.3	AA_0.6	CK	AA_0.3	AA_0.6
Ala	21.115	31.63	33.801	29.995	42.639	40.544
Glu	28.093	35.974	0.253	23.778	26.698	43.501
Gln	660.725	594.785	648.53	472.735	559.055	553.67
Tyr	37.847	44.372	64.462	13.641	9.937	26.319
Arg	121.083	125.461	125.512	44.709	60.074	60.532
Val	81.479	83.763	116.249	43.038	58.853	63.854
Phe	47.764	52.596	143.385	21.087	19.243	27.005
Ile	29.651	35.947	53.781	19.142	20.508	31.371
Thr	15.074	23.565	21.844	15.021	26.02	23.875
Met	32.723	37.512	83.406	20.518	24.828	30.077
Asn	263.491	234.975	226.144	124.503	160.128	229.772
Ser	15.738	18.101	23.223	12.216	17.583	19.748
Asp	29	38.192	33.144	20.929	26.944	32.303
Citr	2.455	3.316	14.812	0.923	0.668	0.909
Pro	77.795	81.099	83.889	63.277	20.879	65.078
Gly	6.566	6.825	12.598	5.248	7.736	7.139
Leu	13.069	14.655	40.839	7.988	9.849	12.427
Trp	24.09	40.875	129.448	18.912	36.165	44.818
Orn	16.895	17.254	59.693	9.689	9.806	7.32
Lys	35.014	38.67	76.811	24.096	28.181	35.831
β-ala	0.179	6.143	0.288	0.14	0.212	0.314
Cys	5.676	6.332	22.486	7.089	8.567	4.049

**Table 2 foods-12-03951-t002:** Volatile substance content in potato tubers under amino acid concentration treatment.

Name	Relvtive Peak Area × 10^2^ (%)					
		D1418			D23	
	0.3_AA	0.6_AA	CK	0.3_AA	0.6_AA	CK
1-Hexanol 2-ethyl-	512,459.5	437,399.5	72,187.5	527,603	345,730	88,660
1-Octanol	199,450	129,537	10,675	185,247	108,486.3	6084
1H-Pyrrole, 1-(2-furanylmethyl)-	2,085,267.5	1,235,397	2650.5	981,915.5	739,987.7	2155
2-(2-Methylpropyl)-3-(1-methylethyl)pyra	16,971,213.5	10,679,824	168,642	15,017,781.5	8,622,082.5	230,322
2-Decenal (E)-	18,663	2777	0	476.5	1084.5	344
2-Dodecenal (E)-	8675.5	18,181	3482.5	47,588	23,083.8	2573
2-Heptanone	54,388	31,966	10,519.5	35,454	25,979.8	7470.5
2-Heptenal (E)	399	106,655	44,725.5	9578.5	53,653	16,703.5
2-Hexenal (E)-	4348	5240	4526.5	16,959	8908.5	0
2-Isoamyl-6-methylpyrazine	405,664.5	347,880	25,621.5	509,848	294,449.8	32,323.5
2-Isobutyl-3-methylpyrazine	10,960,588	7,320,303.5	81,422.5	886,6978	5,422,901.3	86,452
2-n-Butyl furan	8080.5	3095	206.5	7959	3753.5	0
2-Nonenal (E)-	630,186.5	540,706	84,725	444,203	356,544.7	87,146.5
2-Octenal (E)-	1132.5	1325.5	5930	525.5	2593.7	2692.5
2-Undecenal	63,375.5	50,024.5	27,003.5	69,184	48,737.3	2105
2,3-Dimethyl-5-ethylpyrazine	1,796,376	1,255,266.5	72,448.5	1,502,808	943,507.7	55,435.5
2,3-Octanedione	27,686	112,349	114,718.5	716	75,927.8	2871.5
2,3,5-Trimethyl-6-(2-methylbutyl)pyrazin	1,451,749.5	1,094,330	15,482	1,911,333.5	1,007,048.5	23,032.5
2,4-Decadienal, (E,E)-	782,613	686,539	425,574.5	480,687.5	530,933.7	19,410.5
2,4-Heptadienal, (E,E)-	22,916	15,054	1824.5	57,702.5	24,860.3	566.5
2,4-Nonadienal, (E,E)-	23,089.5	15,059.5	31,206	30,868	25,711.2	0
2,4-Undecadienal	4847	6102	5284.5	5348	5578.2	226.5
2,6-Nonadienal (E,E)-	23,708.5	18,828	34,025.5	12,392	21,748.5	27,418
3-Heptanone,2-methyl-	5,3048,701.5	54,602,032.5	11,231,486	40,713,983.5	35,515,834	15,115,644.5
3-Octen-2-one	24,003.5	18,637	25,384.5	15,405.5	19,809	469.5
3,5-Dimethyl-2-isobutylpyrazine	667,238.5	427,661	8776	460,271	298,902.7	13,802
3,5-Octadien-2-ol	32,544.5	28,844.5	1160.5	20,113.5	16,706.2	0
4,5-Dimethyl-2-isopropyloxazole	687,367.5	467,580.5	2361.5	676,230.5	382,057.5	1032.5
5-Hepten-2-one 6-methyl-	246,326	491,362	23,806	374,973.5	296,713.8	45,391
5H-5-Methyl-6,7-dihydrocyclopentapyrazin	1,663,741.5	1,063,249	6034	1,446,688	838,657	9155.5
Benzaldehyde	4,484,489.5	3,147,999.5	738,544	6,176,531	3,354,358.2	1,127,359.5
Benzeneacetaldehyde	6,364,881	6,172,831	905,105	13,624,411	6,900,782.3	1,609,060
Butanal 2-methyl	567,638.5	617,784	49,920	1,550,692.5	739,465.5	38,968
Butanal 3-methyl	10,959	20,286	8595	20,454	16,445	4486.5
Decanal	472,853	408,414.5	168,911.5	496,179.5	357,835.2	122,289
Dibutyl phthalate	14,082.5	17,288.5	10,530.5	14,232.5	14,017.2	9605.5
Dimethyl trisulfide	1,479,351	134,992.5	588,334.5	188,881.5	304,069.5	168,748.5
Disulfide, dimethyl	3,591,356	568,467	191,960	1,808,566	856,331	408,833
Ethanone, 1-(1H-pyrrol-2-yl)-	7,890,385	5,406,293	4801	43,41,755.5	3,250,949.8	13,607.5
Furan 2-methyl	155,971.5	46,441	2365	140,383	63,063	3873
Furan 2-pentyl-	522,208	268,804.5	152,475.5	554,196	325,158.7	125,227
Furfural	6,929,990	351,426.5	4386	2143,526.5	833,113	2407.5
Furfural, 5-methyl-	6,200,283	554,513.5	1172.5	3,605,694.5	1,387,126.8	0
Heptanal	351,220	184,604	166,322	319,639.5	223,521.8	97,844.5
Hexanal	1,267,477.5	817,366	2,308,175.5	2,344,020.5	1,823,187.3	411,623
Methional	7,324,469	2,274,775.5	1,803,119	7,580,234.5	3,886,043	2,447,959
Naphthalene	16,663	12,785.5	0	10,423.5	7736.3	579
Naphthalene 2-methyl-	23,123	4897.5	0	12,879.5	5925.7	0
Nonanal	94,512.5	29,747	0	31,915	20,554	0
Octanal	686,610	385,907.5	139,016	568,157.5	364,360.3	93,030.5
Oxazole, trimethyl-	1,360,370.5	705,664.5	0	535,102	413,588.8	0
Phenol 2-methoxy-	233,424.5	442,647.5	145,789.5	3,340,033.5	1,309,490.2	178,036
Propanal 2-methyl-	2,906,580.5	2,133,307.5	123,169	13,17,808	1,191,428.2	70,286
Pyrazine,	929,673.5	870,579	1047.5	875,061.5	582,229.3	762.5
Pyrazine, 2-ethenyl-6-methyl-	9,281,106	10,237,992.5	120,384.5	7,146,457	5,834,944.7	172,323.5
pyrazine, 2-ethyl-3-methyl-	25,283,235	25,027,548	198,264	9,781,194	11,669,002	133,329.5
pyrazine, 2-ethyl-5-methyl-	636,042	1,134,905.5	7088	705,381	615,791.5	8710
Pyrazine, 2-ethyl-6-methyl-	33,490,667.5	50,884,877.5	137,283.5	32,411,201	27,811,120.7	128,274.5
Pyrazine, 2,3-diethyl-5-methyl-	1,719,778	1,309,362	40,042.5	891,565	746,989.8	30,240
Pyrazine, 2,3-dimethyl	6,364,209	4,528,647.5	102,383.5	5,922,310.5	3,517,780.5	1548.5
Pyrazine, 2,3-dimethyl-5-(2-methylpropyl	1,950,925.5	1,423,921	16,650	1,650,089	1,030,220	23,849.5
Pyrazine, 2,5-diethyl-	2,750,524	2,302,242.5	107,681	2,528,630	1,646,184.5	107,905.5
Pyrazine, 2,5-dimethyl-	233,282	179,421	4824	238,776.5	141,007.2	3889
Pyrazine, 2,5-dimethyl-3-(2-methylpropyl	5,715,465	5,773,194.5	96,171.5	7,771,879.5	4,547,081.8	121,422
Pyrazine, 2,5-dimethyl-3-(3-methylbutyl)	2,970,391	2,225,034.5	31,534	3,722,482.5	1,993,017	38,555
Pyrazine, 2,6-dimethyl	57,821,792.5	31,092,731.5	27,078.5	59,054,175	30,057,995	23,567.5
Pyrazine, 3-ethyl-2,5-dimethyl	102,969,897	85,418,501	1,780,924.5	59,578,984	48,926,136.5	1,266,235.5
Pyrazine, 3,5-diethyl-2-methyl-	20,807,908	18,040,734.5	318,802.5	12,248,286.5	10,202,607.8	258,228
Pyrazine, ethenyl-	1,797,855.5	2,030,248	6541.5	1,829,591.5	1,288,793.7	4904.5
Pyrazine, ethyl-	698,437	650,239	8910	695,049.5	451,399.5	9952
Pyrazine, methyl-	54,750,739	45,196,613.5	5066.5	47,333,874.5	30,845,184.8	0
pyrazine, trimethyl-	15,060,797.5	3,151,401	32,095	20,960,252	8,047,916	15,097
Pyrrole	1,957,153.5	558,749.5	2571	1,343,357.5	634,892.7	9786
Undecanal	132,075	20,445.5	11,672	81,748	37,955.2	14,976

**Table 3 foods-12-03951-t003:** Aroma threshold and ROAV values of volatile flavor compounds in potatoes under amino acid water-soluble fertilizer treatment.

Name	Threshold (μg·kg^−1^)	Aroma Characteristic	ROAV					
				D1418			D23	
			0.6_AA	0.3_AA	CK	0.6_AA	0.3_AA	CK
Ethanone, 1-(1H-pyrrol-2-yl)-	114.39	Bread aroma	<0.01	<0.01	<0.01	<0.01	<0.01	<0.01
Pyrazine, 3-ethyl-2,5-dimethyl	7.257	Baking odor	0.1	0.1	<0.01	<0.01	<0.01	<0.01
1H-Pyrrole, 1-(2-furanylmethyl)-	0.1	Roasted, hazelnut, coffee like aroma	0.1	0.1	<0.01	<0.01	<0.01	<0.01
Phenol 2-methoxy-	0.02	Special fragrance	<0.01	0.1	0.6	0.4	0.8	0.6
Pyrazine, 2,3-diethyl-5-methyl-	0.001	Coffee, nuts	6.3	6	3.1	4.9	4	2.2
pyrazine, 2-ethyl-3-methyl-	0.315	Roasted flavor	0.3	0.4	<0.01	0.2	0.1	<0.01
Pyrazine, 3,5-diethyl-2-methyl-	-	Roasted flavor	-	-	-	-	-	-
2,6-Nonadienal (E,E)-	0.001	Melon, Fatty	0.2	0.2	5.3	0.3	0.1	4
2-Isobutyl-3-methylpyrazine	0.002	Roasted flavor	20.2	16.7	3.2	18	20	3.1
2,4-Heptadienal, (E,E)-	0.1	Fat, fruits, spices	<0.01	<0.01	<0.01	<0.01	<0.01	<0.01
Benzaldehyde	350	Almond flavor	<0.01	<0.01	<0.01	<0.01	<0.01	<0.01
Benzeneacetaldehyde	4	Almond, cherry	<0.01	<0.01	<0.01	<0.01	<0.01	<0.01
Furfural	6.474	Almond flavor	<0.01	<0.01	<0.01	<0.01	<0.01	<0.01
Pyrazine, 2-ethenyl-6-methyl-	-	Roasted flavor	-	-	-	-	-	-

Note: ‘-’ represent not measured.

## Data Availability

Data are contained within the article.

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
