# Peer review of "The Impact of the Foliar Application of Amino Acid Aqueous Fertilizer on the Flavor of Potato Tubers"

_foods, 2023, doi:10.3390/foods12213951_

Round 1

Reviewer 1 Report

1. Reference 1 dressed on the Planting and exporting of fresh potato. It did not say eat fresh potato. Please recheck and make necessary correction.

2. In line 38, do you mean RNA is a flavor precursor? May be nucleotides is better.

3. English in this manuscript need extensive editing.

4. Line 57, “no reports have been found” what do you mean?

5. In line 71 “The fertilizer used for the experiment is an amino acid mixed foliar fertilizer  with a purity >99%” How to define a purity >99% for a mixed amino acid. And what amino acid? Is it only contain one amino acid? Or amino acids? It is very confusing.    

6. Line 100, please dressed the headspace method. By HS sampling? By HS SMPE? Please describe the method briefly, as it was a very important part of the data come from.

7. It is suggested to put tables of the amino acid contents and volatile components in all groups. There are only heatmaps. It is hard to tell how much has it increased?

8. Line 133, how the potato be air fried? Please describe in the method.

English in this manuscript need extensive editing.

Reviewer 2 Report

The manuscript has a similarity index of about 17 % which needs to be reduced by the authors as per the publisher needs.

Lane 89-91: The authors have mentioned that a 1.1 meter protective row of plants were established around the experimental field, wih 4 repititons for each treatment. They need to specify wether these 4 plants were also considered for statistical analysis in the materials and methods section.

Lane 121-123: The mechanism by which Asn and Gln were lowered in 'Dianshu 23' variety with foliar application of amino acid fertilizers can be discussed in a better way here.

The authors are recommended to make use of english language editing service providers for improving the language of the paper.

Reviewer 3 Report

Dear Editor, in the manuscript Foods-26485405 authors evaluated the effects of 0.3 and 0.6 % fertilizer (containing aminoacids) by foliar spray trials on free amino acid content and volatile compounds responsible for tuber flavour in tubers of two potato cultivars, ‘Dianshu 1418’ and ‘Dianshu 23’. The manuscript provides some new and interesting information and need some major revisions:

- Line 72: The amino acid composition of the fertilizer should be addressed.

- Material and method section should be improved by adding information about the procedures for amino acids add volatile compounds quantification, as well as for the correlation test used.

- Line 104: Full name for all the abbreviations used in the formula should be addressed.

- Line 123: If 0.6 % treatment had more effects than 0.3 % treatment, higher concentration of fertilizer than 0.6 % would probably lead to higher effects. Thus, without testing the effect of higher concentrations, it cannot be concluded that 0.6 % was the best concentration. Thus, as commented in lines 197-198 more experiments are needed which would aid to increase the merit of the manuscript.

- Figure 3: It should be added for what variety are these data.

- Figure 4: The same as above.

- Line 202: According to information in line 192, in this Figure only data of ‘Dianshu 23’ are shown. Data of the other cultivar should be provided.

- Line 213: According to information in lines 144-146, 0.3 % treatment had higher impact on volatiles than 0.6 % and in this statement the contrary is claimed.

- Lines 236-245: This information and Figure 5 should be moved to Results section.

- The following previous paper could be useful for discussion section:

Xu, Y., Meng, D., Liu, H.-J., Chen, L., Shi, C.-Y. 2020. Effect of potassium fertilizer on nutritional components of sweet potato storage roots and its relationship with roasting flavor. Journal of Plant Nutrition and Fertilizers, 26(10), pp. 1758–1767.

- References should be written according to the journal format. For instance, use abbreviated journal name.

- Line 348: Check this reference because it seems that it is not correctly written.

- References number 17 and 34 seems to be incomplete.

Round 2

Reviewer 3 Report

The comments and suggestions about the original manuscript have been addressed in the revision version of the manuscript, so that it could be suitable for publication.